# Tradeoffs explain scaling, sex differences, and seasonal oscillations in the remarkable weapons of snapping shrimp (*Alpheus spp.*)

**Jason P Dinh\*, SN Patek**

Department of Biology, Duke University, Durham, United States

**Abstract** Evolutionary theory suggests that individuals should express costly traits at a magnitude that optimizes the trait bearer's cost-benefit difference. Trait expression varies across a species because costs and benefits vary among individuals. For example, if large individuals pay lower costs than small individuals, then larger individuals should reach optimal cost-benefit differences at greater trait magnitudes. Using the cavitation-shooting weapons found in the big claws of male and female snapping shrimp, we test whether size- and sex-dependent expenditures explain scaling and sex differences in weapon size. We found that males and females from three snapping shrimp species (Alpheus heterochaelis, Alpheus angulosus, and Alpheus estuariensis) show patterns consistent with tradeoffs between weapon and abdomen size. For male A. heterochaelis, the species for which we had the greatest statistical power, smaller individuals showed steeper tradeoffs. Our extensive dataset in A. heterochaelis also included data about pairing, breeding season, and egg clutch size. Therefore, we could test for reproductive tradeoffs and benefits in this species. Female A. heterochaelis exhibited tradeoffs between weapon size and egg count, average egg volume, and total egg mass volume. For average egg volume, smaller females exhibited steeper tradeoffs. Furthermore, in males but not females, large weapons were positively correlated with the probability of being paired and the relative size of their pair mates. In conclusion, we identified size-dependent tradeoffs that could underlie reliable scaling of costly traits. Furthermore, weapons are especially beneficial to males and burdensome to females, which could explain why males have larger weapons than females.

\*For correspondence:
jasonpdinh@gmail.com

Competing interest: The authors declare that no competing interests exist.

## Editor's evaluation

This study on snapping shrimp morphological weaponry presents important findings on trade-offs in investment in costly weaponry traits as related to body size and reproduction. Convincing evidence is based on the collection of an exceptional number of fields samples, the inclusion of three shrimp species, and the measurement of numerous morphological and behavioral traits. The evidence shows that there are size-dependent trade-offs. Further, males and females differ in weapon investment, as weapons are especially beneficial to males but especially expensive for females. The findings will be of broad interest to evolutionary biologists and researchers working in the field of animal behavior.

## Introduction

Weapons, ornaments, and other secondary sexual traits often scale with the trait-bearer's quality. Larger weapons can better deter or damage competitors, and more intense ornaments can better

**eLife digest** From deer antlers to crab claws, weapons are some of the most elaborate and enormous structures in the animal kingdom. Within a species, weapon size generally increases with the size and condition of an individual, and those with larger weapons are usually better at fending off more diminutive competitors.

Although it may seem desirable for all individuals to have large weapons, size varies greatly within a species. The 'handicap principle' proposes that the cost of bearing a weapon dictates the variation in weapon size. Smaller or less fit individuals pay more for weapons than larger or fitter animals, so smaller individuals tend to grow smaller weapons. Although popular, only a handful of studies have demonstrated experimental evidence that supports this theory.

To test the handicap principle, Dinh and Patek studied a group of crustaceans known as snapping shrimp. Each shrimp has one enlarged claw that it uses as a weapon to fire imploding vapor bubbles at opponents during fights. Larger snapping shrimp have bigger enlarged claws and tend to win more contests. Males also have larger weapons than females, and this sex difference is amplified during the breeding season.

Dinh and Patek studied weapon size in several species of snapping shrimp. Measurements showed that after controlling for body size, individuals with larger weapons had smaller abdomens, suggesting there is a tradeoff between weapon size and abdomen size. Furthermore, small males exhibited the steepest tradeoff, in line with the handicap principle.

Snapping shrimp also showed sex-specific costs and benefits. After controlling for body size, females with larger weapons produced fewer and smaller eggs, while males with larger weapons were more likely to be paired with females and generally paired with larger females. This suggests that weapons are particularly burdensome to female shrimp and particularly beneficial to males, especially during the breeding season.

These findings provide elusive evidence for the handicap principle and extend the theory to explain sex and seasonal differences in the size of snapping shrimp weapons. More broadly, the findings highlight the value of studying both male and female animal weapons when, historically, the focus has been on male weaponry.

---

attract mates. By first approximation, one might expect that all individuals should express these traits to arbitrarily high magnitudes because greater expression yields fitness benefits. However, fitness costs and physical limitations ensure that traits are expressed honestly instead of arbitrarily (reviewed in *Searcy and Nowicki, 2005*). Despite decades of research, the costs that maintain reliable scaling relationships remain hotly debated.

One hypothesis called the handicap principle suggests that sexual traits are costly, and these costs ensure that trait expression is not arbitrary. Costly traits lower fitness by reducing survival (*Kotiaho et al., 1998*; *Moller and de Lope, 1994*; *Mappes et al., 1996*) or reproduction (*Cavender et al., 2021*; *Joseph et al., 2018*; *Moczek and Nijhout, 2004*; *Somjee et al., 2018*). Individuals should therefore express traits at a level that maximizes their benefits relative to their unit of cost (*Grafen, 1990a*; *Grafen, 1990b*; *Nur and Hasson, 1984*; *Zahavi, 1977*). For example, the handicap principle posits that sexually selected traits scale with quality because low-quality individuals pay more for, or benefit less from, costly traits compared to high-quality individuals. These differential costs set the optimal trait expression at a lower value for lower-quality individuals compared to higher-quality ones (*Grafen, 1990a*; *Grafen, 1990b*; *Nur and Hasson, 1984*; *Zahavi, 1977*). Even though this is a widely accepted explanation for the honest scaling of sexual traits, empirical evidence is scarce (*Kotiaho, 2001*; *Penn and Számadó, 2020*).

In addition to scaling relationships, costly traits can also differ depending on sex and season. For example, some secondary sexual traits are expressed in both sexes but at greater magnitudes in males than females (*Heuring and Hughes, 2019*; *Nolazco et al., 2022*). Moreover, costly traits might be expressed more intensely during the breeding season compared to the nonbreeding season, such as the annual shedding and regeneration of deer antlers (*Brockes et al., 2004*; *Clements et al., 2010*; *Price et al., 2005*). Snapping shrimp offer a particularly tractable system with which to test these classic questions about scaling, sex, and seasonality in the expression of costly traits.

Snapping shrimp live in size-assortative male-female pairs. Both males and females in the pair defend territory, maintain shelter, and forage (*Hughes et al., 2014*; *Mathews, 2002a*). Size-matched pairs form via intraspecific contests and intersexual mate choice, but the exact dynamics of pair formation differ depending on the species (*Heuring and Hughes, 2020*; *Rahman, 2002*; *Rahman et al., 2004*). The eggs in a female's clutch are sired predominantly by the male in the size-matched pair; in other words, extra-pair paternity is rare (*Mathews, 2007*). Furthermore, egg clutch size is a function of female body length, while all reproductively active males can fertilize even the most bountiful of egg clutches (*Knowlton, 1980*). Female snapping shrimp are only reproductively receptive for several hours after each molt, which occurs once every 16–20 days (*Govind et al., 1986*; *Knowlton, 1980*; *Mathews, 2002b*; *Rahman et al., 2003*). Meanwhile, males are not limited to this molt-related breeding cycle. The estimated longevity for snapping shrimp ranges from 13 to 16 months (*Costa-Souza et al., 2018*; *Mossolin et al., 2006*).

Individuals of both sexes bear one enlarged claw that they use as weapon during fights with same-sex conspecifics (*Nolan and Salmon, 1970*). They assess weapons as visual signals (*Hughes, 1996*) and use them as armament to injure or damage opponents (*Dinh et al., 2020*; *Dinh and Patek, 2023*; *Kingston et al., 2022*). Snapping shrimp use latch-mediated spring actuation to produce powerful strikes (*Kaji et al., 2018*; *Longo et al., 2019*; *Longo et al., 2023*; *Patek and Longo, 2018*). They cock their claws open and use muscles to load an elastic mechanism comprised of the flexing exoskeleton and stretching apodemes (*Longo et al., 2023*). They unlatch the claw to quickly release elastic energy, driving the dactyl shut in as little time as 0.36 milliseconds (*Dinh and Patek, 2023*). Upon closure, a tooth-shaped protrusion in the dactyl inserts into a cavity in the propodus, which generates a high-velocity water jet that vaporizes the trailing region of water. This vapor bubble, known as a cavitation bubble, collapses and produces pressures that are audible to the human ear as a 'snap' (*Kaji et al., 2018*; *Lohse et al., 2001*; *Versluis et al., 2000*). Snapping shrimp fire snaps at opponents during contests (*Dinh et al., 2020*; *Dinh and Patek, 2023*; *Nolan and Salmon, 1970*). The pressure of the cavitation bubble collapse can cause neurotrauma to the opponent, so snapping shrimp have evolved shock-absorbing helmets called orbital hoods to dampen the blows (*Kingston et al., 2022*).

Individuals with larger weapons produce longer-lasting cavitation bubbles, greater pressures, and have greater offensive capacity (*Dinh and Patek, 2023*). They also tend to win contests (*Dinh et al., 2020*; *Dinh and Patek, 2023*). Yet, snapping shrimp do not grow weapons to arbitrary sizes. Instead, they vary along three axes: (1) larger individuals have larger weapons, (2) at any given body size, males have larger weapons than females, and (3) the sex difference amplifies during the summer breeding season (*Heuring and Hughes, 2019*). Therefore, the costs and benefits of weapon size can be examined across these three axes: body size, sex, and breeding season.

We test if snapping shrimp face tradeoffs that scale with the condition as predicted by the handicap principle. Then, we test the hypothesis that sex and seasonal differences in weaponry arise from sex-specific costs and benefits in alpheid snapping shrimp. We did not measure fitness and, therefore, refrain from using the term 'costs' when referring to our data. Instead, we use the term expenditure to represent tradeoffs that could cascade to fitness costs (*Kotiaho, 2001*).

## Results
### Morphology

To identify weapon expenditures that vary with size as predicted by the handicap principle, we tested if snapping shrimp individuals bearing large weapons sacrificed resources from the abdomen (the muscular segmented region of the body used for swimming) (*Arnott et al., 1998*; *Hunyadi et al., 2020*). Reduced abdomen size could lower fitness through reduced survival, given that abdomen length is positively correlated with predator escape velocity in other benthic decapod crustaceans (*Hunyadi et al., 2020*). Snapping shrimp with smaller abdomens could therefore be more vulnerable to predation. Furthermore, female snapping shrimp hold eggs underneath their abdomen, and reduced abdomen size could constrain maximum egg clutch volume. Thus, we tested whether snapping claws exhibit a morphological tradeoff with abdomen size, and whether this expenditure increases as body size decreases.

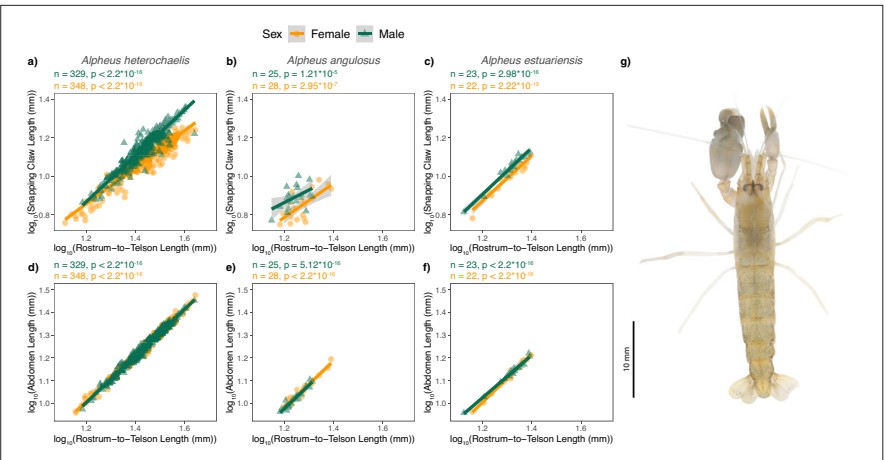

**Figure 1.** Snapping claw length and abdomen length increased with rostrum-to-telson length across the three alpheid species. Residuals from these lines were used to test for weapon expenditures and tradeoffs in subsequent analyses. Shaded regions represent 95% confidence intervals for linear regressions. A scaled dorsal view of an *Alpheus angulosus* individual is shown in panel g (distal toward the top of the page; the left claw is the snapping claw). Slopes of each scaling relationship are presented in *Supplementary file 1*, Table 1. F-test sample sizes and p-values are shown above each graph.

The allometric slope of snapping claw scaling differed significantly between sexes for *Alpheus heterochaelis* and *Alpheus angulosus* but not for *Alpheus estuariensis* (the species for which we had the smallest sample size) (*Figure 1*). Scaling slopes and 95% confidence intervals are presented in *Supplementary file 1*, Table 1. We used the residuals from scaling relationships for the snapping claw and abdomens to test for morphological tradeoffs between weapons and abdomens (see Materials and methods).

As predicted, weapons with greater snapping claw residuals exhibited tradeoffs with abdomen length. Snapping claw residuals and abdomen residuals were negatively correlated in both sexes and for all three species (*Figure 2*; *Supplementary file 1*, Tables 2-4). We tested if this tradeoff was size-dependent in *A. heterochaelis* — the species for which we had the largest sample size and greatest statistical power. For males, as predicted, individuals with smaller carapace lengths had steeper tradeoff slopes compared to those with larger carapace lengths (interaction p-value = 0.002; *Figure 3*; *Supplementary file 1*, Table 5). By contrast, we found no evidence of size-dependent slopes for female weapons (interaction p-value = 0.93; *Supplementary file 1*, Table 6).

## Kinematics

Larger weapons produce longer-lasting cavitation bubbles and greater pressures (*Dinh and Patek, 2023*). However, individuals that grow larger weapons than predicted by snapping claw scaling relationships do so using less muscle and more exoskeleton (*Dinh, 2022*). Reducing the amount of muscle in the claw may hinder elastic loading and snap production. Therefore, we tested if growing weapons larger than predicted by the weapon size scaling relationships reduced the average angular velocity of the snapping claw, cavitation bubble duration, or pressure of the snap. We predicted that this tradeoff would be steepest in the smallest males as predicted by the handicap principle.

Surprisingly, weapon residuals did not affect any measured snap parameter in *A. heterochaelis* males or females: Neither weapon residual nor its interaction with claw mass were significant predictors of $\log_{10}$(average angular velocity), $\log_{10}$(bubble duration), or sound pressure level (*Supplementary file 1*, Table 7-12).

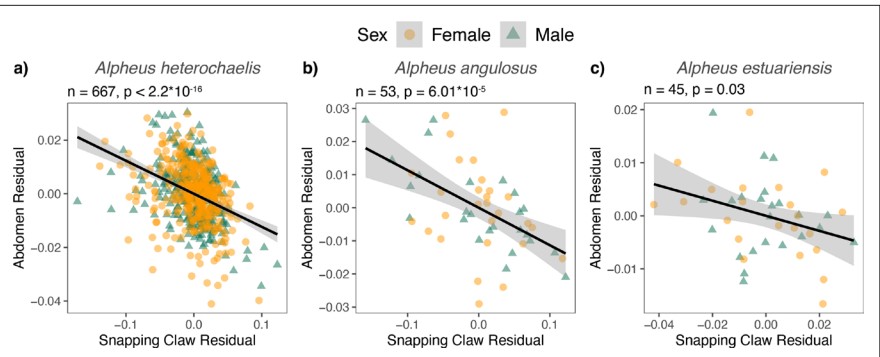

**Figure 2.** In all three analyzed species, there was a tradeoff between snapping claw residuals and abdomen residuals. Individuals with greater snapping claw residuals had lower abdomen residuals in (**a**) *Alpheus heterochaelis*, (**b**) *Alpheus angulosus*, and (**c**) *Alpheus estuariensis*. Regressions were calculated from both sexes because sex and the sex × snapping claw residual interaction were not significant predictors in any model. Shaded regions represent 95% confidence intervals for linear regressions. F-test sample sizes and p-values are shown above each graph.

## Reproductive tradeoffs

To determine if female-specific expenditures explain why females have smaller proportional weapon sizes than males, we tested for tradeoffs between female weaponry and egg production. Analogous tradeoffs between primary and secondary sexual characteristics arise for males in taxa as diverse as narwhals and dobsonflies (*Dines et al., 2015*; *Liu et al., 2015*; *Simmons et al., 2017*). In snapping shrimp, females bear the entire burden of egg production (*Knowlton, 1980*). Therefore, resources allocated to costly traits like weaponry should reduce the allotment invested in primary reproduction. We used the same residual tradeoffs approach that was used to test for morphological tradeoffs between weapons and abdomens.

For female *A. heterochaelis*, weapon residuals had egg production tradeoffs. Weapon residuals were negatively correlated with egg mass volume residuals, average egg volume, and egg count residuals (*Figure 4*; *Supplementary file 1*, Table 13-15). Tradeoffs for egg count residuals and egg mass volume residuals were not size-dependent ($p_{interaction}$ = 0.223 and $p_{interaction}$ = 0.483, respectively). However, average egg volume tradeoffs were steeper for females with smaller carapace lengths compared to those with larger carapace lengths (interaction term *t*-test: b=1.241, SE = 0.538, t=2.306, p=0.028) (*Figure 5*; *Supplementary file 1*, Table 15).

## Pairing

If males benefit more from large weaponry than females, then that benefit could also contribute to the sex differences in weaponry. Therefore, we tested if males with large weaponry benefited through improved pairing success. Snapping shrimp form size-assortative pairs (*Mathews, 2002b*; *Nolan and Salmon, 1970*). We tested whether large weapons improved the likelihood of pairing and whether individuals with large weapons paired with relatively larger mates. If either of these pairing advantages disproportionately benefits males, then this could explain why males have larger weapons than females.

In *A. heterochaelis*, paired males had significantly greater weapon residuals compared to unpaired males (*t*-test: n=233, p=0.000299), but there was no significant difference for females (*t*-test: n=253, = 0.56) (*Figure 6*; *Supplementary file 1*, Table 16).

For males, the probability of being paired increased as snapping claw residual increased (n=233, b=16.879, SE = 5.652, z=2.986, p=0.00345), but there was no significant relationship with carapace length (p=0.104; *Supplementary file 1*, Table 17). By contrast, for females, the probability of being paired increased as carapace length increased (n=253, b=0.574, SE = 0.142, z=4.034, p=3.72×10$^{-5}$) but there was no significant relationship with snapping claw residual (p=0.487; *Supplementary file 1*, Table 18; *Figure 6*).

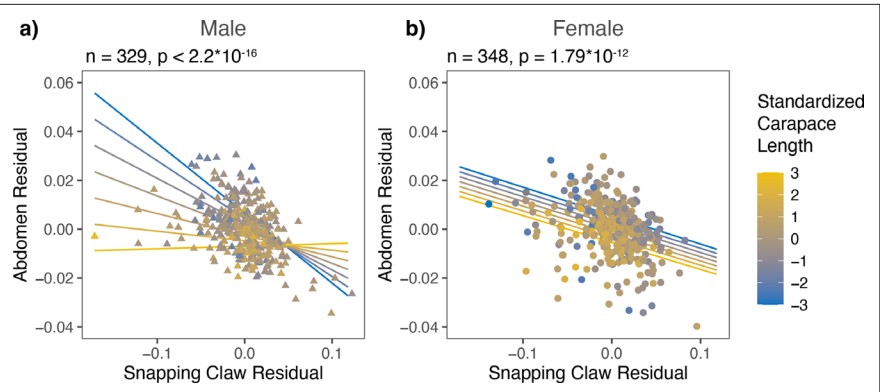

**Figure 3.** The tradeoff between snapping claws and abdomens varied by size in *Alpheus heterochaelis* males. The tradeoff between snapping claw residuals and abdomen residuals was steepest for the smallest individuals in *Alpheus heterochaelis* males (**a**) but not females (**b**). Lines represent model predictions for standardized carapace lengths of −3, −2, −1, 0, 1, 2, and 3. A standardized carapace length of 0 represents an individual with the mean carapace length, and each increment of 1 represents one standard deviation. F-test sample sizes and p-values are shown above each graph. The interaction term was significant for males (*t*-test, n=329, p=0.00209) but not for females (*t*-test, n=348, p=0.932).

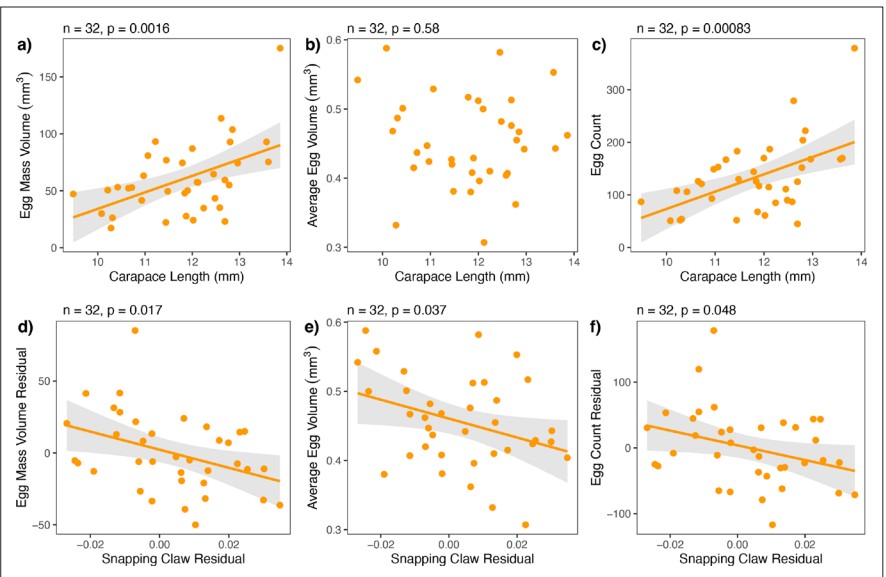

**Figure 4.** *Alpheus heterochaelis* females exhibited tradeoffs between weapon size and egg mass volume, average egg volume, and egg count. As carapace length increased, (**a**) egg mass volume increased, (**b**) average egg volume remained constant, and (**c**) egg count increased. As snapping claw residuals increased, (**d**) egg mass volume residuals decreased, (**e**) average egg volume decreased, and (**f**) egg count residual decreased. F-test sample size and p-values are shown above each graph.

For paired males, as weapon residuals increased, the relative rostrum-to-telson lengths of their pair mates also increased (linear model F-test, n=111, p=0.00467). However, there was no significant trend in females (linear model F-test, n=111, p=0.0649) (*Figure 6*; *Supplementary file 1*, Table 19, 20).

## Seasonal trends

Because the benefits of being paired and the costs of egg production are most salient during the breeding season, we expected investment into different morphologies to change as the costs and benefits do. Specifically, we predicted that snapping claw residuals would be greater for males during the breeding season. Meanwhile, we predicted that abdomen residuals for both sexes would decrease

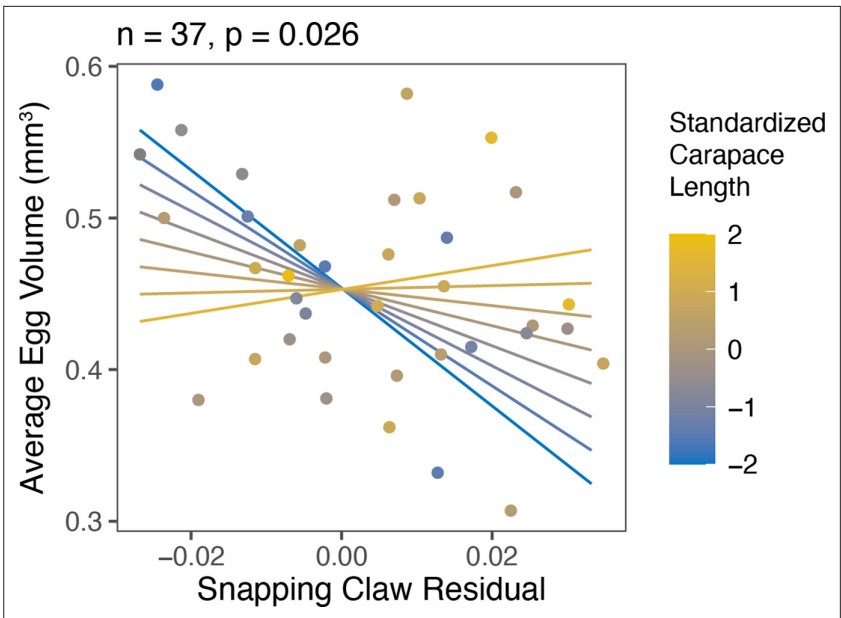

**Figure 5.** Smaller *Alpheus heterochaelis* females (blue) exhibited steeper tradeoffs between snapping claw residuals and average egg volume compared to larger females (yellow). Lines represent model predictions for standardized carapace lengths of −2, −1.5, −1, −0.5, 0, 0.5, 1, and 1.5. A standardized carapace length of 0 represents an individual with the mean carapace length, and each increment of 1 represents one standard deviation.

during the breeding season as males invest more into their weapons and as females invest more into their eggs.

Abdomen residuals were reduced in male *A. heterochaelis* during the breeding season compared to the non-breeding season, whereas females exhibited a marginally nonsignificant but parallel seasonal shift (*t*-test: n=348, p=0.06) (*Figure 7*). Meanwhile, snapping claw residuals were elevated in males during the breeding season compared to the non-breeding season, whereas females exhibited no significant seasonal shift (*Figure 7*).

Furthermore, the scaling slope for female snapping claws became less steep during the breeding season (interaction term *t*-test: n=348, b=−0.183, p=0.000838). There was no such seasonal shift in allometry for males (interaction term *t*-test: n=329, p=0.233). After the nonsignificant interaction term was removed from the male model, there was a significant increase in snapping claw lengths across all rostrum-to-telson lengths (*t*-test, n=329, b=0.023, = 5.62×10$^{-6}$) (*Figure 7*; *Supplementary file 1*, Table 22-24).

## Discussion

Evolutionary theory suggests that individuals express costly traits like weapons and ornaments at an optimal magnitude that maximizes the cost-benefit difference. Because individuals differ in the costs they pay and the benefits they reap, trait expression varies in systematic and predictable ways across the population (*Grafen, 1990a*; *Grafen, 1990b*; *Nur and Hasson, 1984*; *Zahavi, 1977*). We found empirical evidence for size-dependent expenditures that could explain reliable scaling of trait expression: The smallest snapping shrimp exhibited the steepest morphological and reproductive tradeoffs. Moreover, we applied the same logic — that costs and benefits differ between individuals and lead to different optimal trait expressions — to explain sex differences in weaponry. Large weaponry is especially burdensome to females which suffer reproductive tradeoffs. Meanwhile, large weaponry benefits males by increasing the probability of being paired and the relative rostrum-to-telson length of their pair mate. These sex-specific implications of weapon investment on reproduction and pairing are vital to fitness because female egg production is the primary determinant of fecundity (*Knowlton, 1980*). Males can boost fitness by pairing with larger females, and females sacrifice fitness

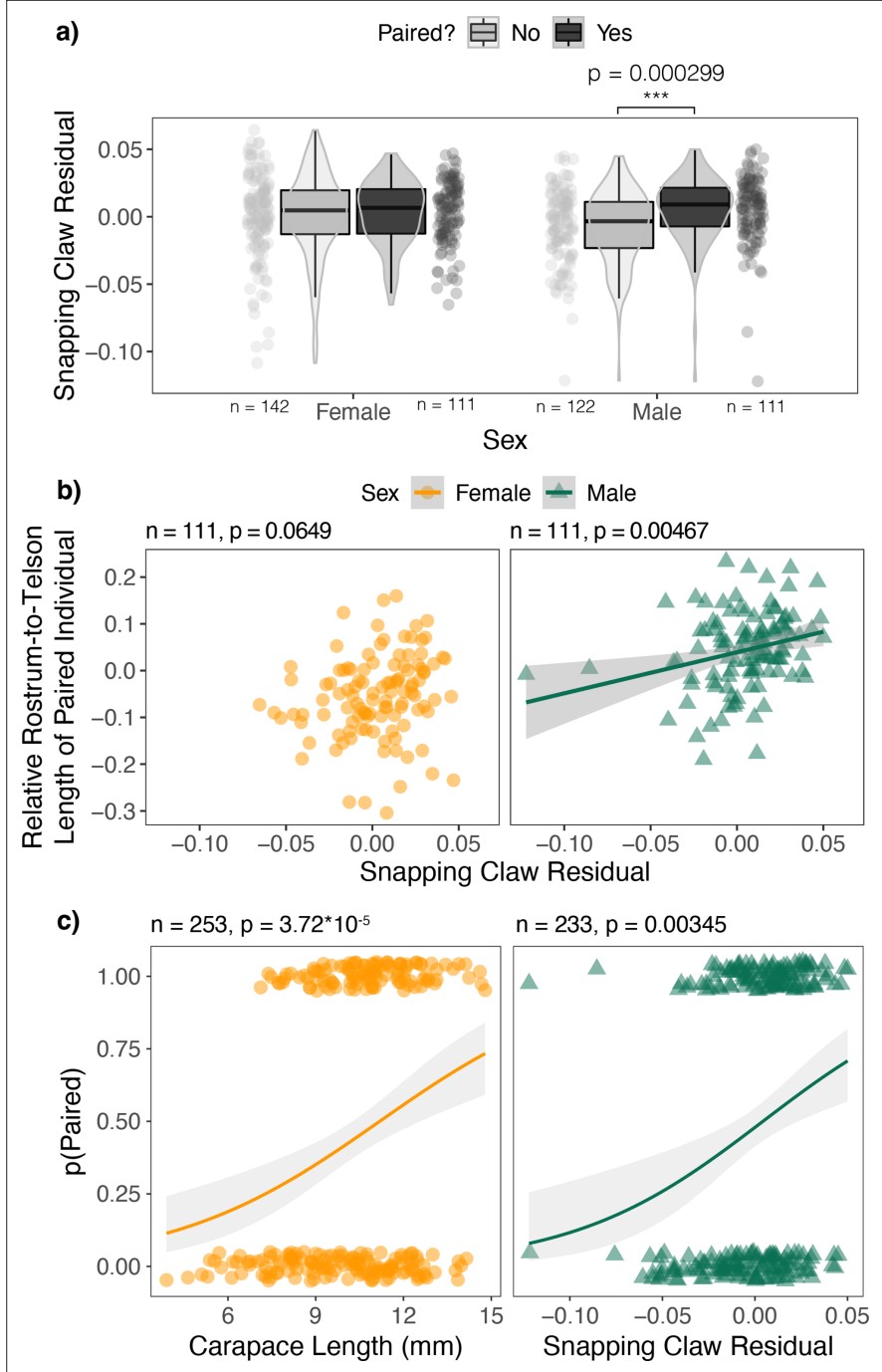

**Figure 6.** Male *Alpheus heterochaelis* benefited from positive snapping claw residuals through pairing in a way that females did not. (**a**) Paired *Alpheus heterochaelis* males had greater snapping claw residuals than unpaired males, but there was no such trend in females. Sample sizes are shown below each jittered dot plot. p-value for the statistically significant *t*-test is shown above the graph. (**b**) Males with more positive residuals paired with relatively larger pair mates, but there was no such trend in females. F-test sample sizes and p-values are shown above each graph. (**c**) The probability of being paired was positively correlated with snapping claw residuals (but not carapace length) for males. Meanwhile, the same probability was correlated with carapace length (but not snapping claw residuals) for females. 1 indicates paired individuals, and 0 indicates unpaired individuals. Z-test sample sizes and p-values are shown above each graph. Shaded regions in regressions are 95% confidence intervals.

by reducing investment into eggs. These sex-specific tradeoffs and benefits can therefore explain why females have smaller proportional weapon sizes compared to males, why this sex difference amplifies during the breeding season, and why female weapon scaling slopes become more shallow during the breeding season when egg production and pairing are at a premium.

For both males and females, individuals with larger weapons had smaller abdomens (*Figure 2*). This was true in all three species of snapping shrimp that we tested. Similar resource allocation tradeoffs between body parts have been shown in other species. For example, several species of dung beetles face tradeoffs between their horns and nearby morphologies such as their eyes, wings, and antennae (*Emlen, 2001*), and reindeer face tradeoffs between antler length and body mass (*Melnycky et al., 2013*). Critically, in our study, we moved beyond simply identifying a tradeoff and demonstrated that different individuals experience tradeoffs to different extents. Specifically, in male *Alpheus heterochaelis*, smaller males exhibited a steeper tradeoff than larger males, indicating a size-dependent expenditure of weaponry (*Figure 3*).

The proportion of the claw made of muscle decreases as the weapon residual increases (*Dinh, 2022*). Therefore, we tested whether weapon residuals were negatively correlated with average angular velocity in the snapping claw, cavitation bubble duration, and snap pressure. We expected weapon residuals to affect these metrics because in other crustaceans, weapon residuals and muscle mass affect weapon function. For example, in some fiddler crabs, regenerated claws never fully recover the muscle mass of the original ones and have reduced pinching force (*Lailvaux et al., 2009*). Similarly, in the same species of fiddler crabs, as weapon residuals increase, pinching force does, too (*Lailvaux et al., 2009*). Surprisingly, in our analysis of snapping shrimp, weapon residuals were not correlated with any of the measured snap parameters.

The expenditures and benefits of growing a large weapon also differed by sex. For ovigerous *A. heterochaelis* females, greater weapon size led to lower egg counts, smaller average egg volume, and lower egg clutch volume (*Figure 4*). In males, tradeoffs between primary reproductive traits and weapons are widespread. For example, male horned scarab beetles (*Onthophagus spp.*) experience tradeoffs between their horns and genitalia, and this tradeoff are most pronounced just as larvae are about to enter their prepupal stage (*Moczek and Nijhout, 2004*; *Simmons and Emlen, 2006*). Similar tradeoffs arise in other insects like coreids (*Miller et al., 2019*; *Somjee et al., 2018*). These tradeoffs are evident in phylogenetic comparative analyses. Across horned scarab beetle species, for example, those with more positively allometric horn slopes have more negatively allometric testes slopes (*Simmons and Emlen, 2006*). Similarly, cetacean species that invest more in sexually dimorphic traits (e.g. narwhal tusks) invest less in testes mass (*Dines et al., 2015*), howler monkeys that have greater hyoid volume have smaller testes mass (*Dunn et al., 2015*), and dobsonflies that have enlarged mandibular weapons invest less into nuptial gifts (*Liu et al., 2015*). Although weapon-reproduction tradeoffs are commonly identified across taxa, the existing studies rarely identify analogous tradeoffs in females, and they rarely, if ever, test for size- or condition-dependence of tradeoffs. Our findings that female snapping shrimp face weapon-reproduction tradeoffs and those reproductive expenditures were size-dependent provide valuable nuances to the existing literature.

In addition, we showed that male *A. heterochaelis* benefited by investing in weaponry through pairing, whereas females did not. In males, weapon residuals were positively correlated with the probability of being paired and the relative body length of their pair mates (*Figure 6*). Females did not exhibit either of these benefits. Male-specific benefits could, therefore, contribute to sex differences in weapon investment.

Egg production is particularly salient to female snapping shrimp because they bear the entire energetic burden of egg production (*Knowlton, 1980*). Likewise, there is an incentive for males to pair with large and fecund females. Therefore, growing a large weapon is particularly burdensome to females and particularly beneficial for males. These reproductive expenditures and benefits could therefore explain why males have larger proportional weapon sizes than females.

The sex-specific expenditures and benefits are also consistent with seasonal oscillations in weaponry. *A. heterochaelis* males had greater weapon residuals during the breeding season compared to the non-breeding season, whereas female weapon residuals remained consistent throughout the year (*Figure 7*). Furthermore, the scaling slope of the snapping claw became more shallow during the breeding season for females. By contrast, males did not show a significant seasonal change in scaling slope, but across the range of body sizes, snapping claw lengths increased during the breeding season

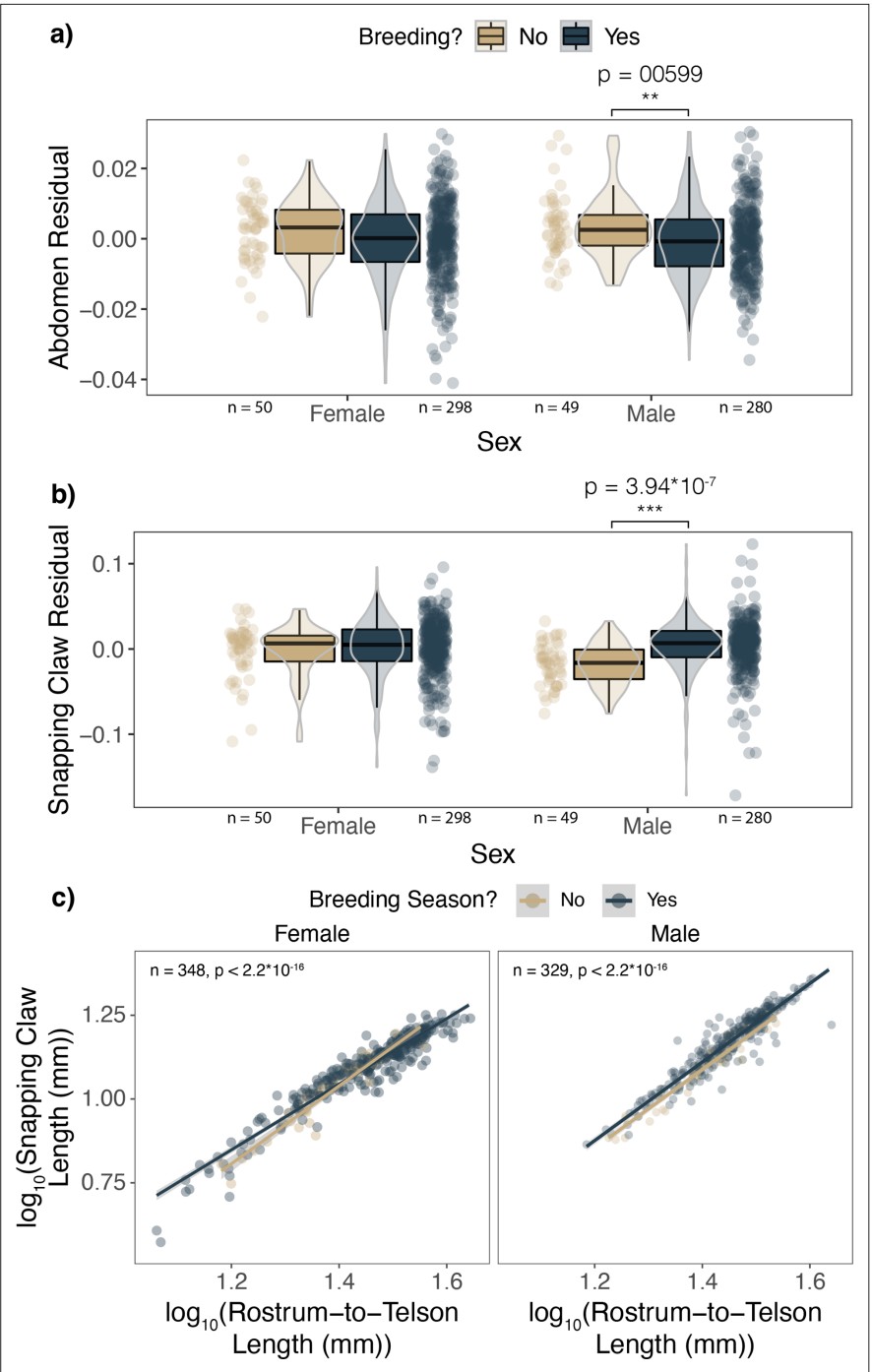

**Figure 7.** Male *Alpheus heterochaelis* shifted investment from their abdomen to their snapping claws during the breeding season, whereas female weapon scaling slopes decreased during the breeding season. During the breeding season, males had (**a**) reduced abdomen residuals and (**b**) increased snapping claw residuals. Females did not exhibit significant morphological shifts. (**c**) Female *Alpheus heterochaelis* scaling slopes were significantly shallower during the breeding season compared to the nonbreeding season. Male scaling slopes did not significantly change seasonally, but during the breeding season, there was an upward shift in snapping claw lengths across all rostrum-to-telson lengths. Shaded regions are 95% confidence intervals. F-test sample sizes and p-values are shown above each graph. \*\*p<0.01 \*\*\*p<0.001.

(*Figure 7*). Concurrently, males had significantly lower abdomen residuals during the breeding season, whereas females exhibited a parallel but marginally nonsignificant decrease in abdomen residuals. Similar trends have been reported in *A. angulosus*, although in that species, females significantly reduce proportional abdomen sizes during the breeding season (*Heuring and Hughes, 2019*). We speculate that males shift investment from their abdomens into weapons during the breeding season because it increases their likelihood of being paired. Female snapping shrimp shift investment from abdomens to eggs, and they do not increase weapon size because they face tradeoffs between eggs and weapons.

Female weapon-egg tradeoffs are analogous to classic examples of male weapon-testes tradeoffs (*Simmons et al., 2017*; *Simmons and Emlen, 2006*). Still, female analogs of this phenomenon are rare (*Miller et al., 2019*). Most likely, the dearth of findings is simply due to insufficient studies of female weaponry. Sex biases in research, such as the misconception that only males fight and only females choose, are common (*Haines et al., 2020*; *Pollo and Kasumovic, 2022*; *Tang-Martínez, 2016*). For example, it is now accepted that female birdsong is widespread, but for centuries, historical research focused almost entirely on males that were presumed to be the only sex to compete for mates (*Odom et al., 2014*; *Odom and Benedict, 2018*; *Riebel et al., 2019*). Like birdsong, female secondary sexual traits, weapons, and competition are not uncommon, and they often serve signaling functions just as they do in males (*Amundsen and Forsgren, 2001*; *LeBas, 2006*; *Nolazco et al., 2022*; *Nordeide, 2002*; *Watson and Simmons, 2010*). Sex-inclusive research on the costs and benefits of these traits would not only redress long-standing omissions from the scientific literature, but comparisons between males and females would also provide empirical tools to understand how costs and benefits govern trait expression within a single species.

In snapping shrimp, large weaponry could have fitness benefits. For example, individuals of both sexes use weapons during contests over mates and territory (*Dinh et al., 2020*; *Dinh and Patek, 2023*; *Hughes et al., 2014*; *Nolan and Salmon, 1970*). Larger weapons produce greater pressure and, therefore, increase offensive capacity during contests (*Dinh and Patek, 2023*). Furthermore, male snapping shrimp with large weapons independent of body size use elevated levels of visual weapon displays, and that seems to affect rival assessment during contests (*Hughes, 2000*). It is also possible that large weaponry could be preferred in mate choice, although that has not been established firmly. In *A. angulosus*, females show a marginally non-significant preference for males with large claws independent of body size during the breeding season (*Heuring and Hughes, 2020*). Meanwhile, females prefer larger males in *A. heterochaelis*, although experiments have not tested whether weapon size independent of body size affects mate choice in this species (*Rahman et al., 2004*). Finally, large weaponry could also help snapping shrimp defend themselves against heterospecific intruders and predators. For example, the snapping shrimp *Alpheus armatus* defends its host anemone from fireworm predation using snaps, which can kill intruders (*McCammon and Brooks, 2014*). Anecdotal evidence suggests a similar symbiotic relationship between nest-defending *A. heterochaelis* mud crabs (*Layman et al., 2003*). If larger weapons are better equipped to fend off predators, then large weaponry could also boost fitness by preventing predation.

Ideally, we would be able to link each of the expenditures and benefits we identified here to a fitness cost (*Kotiaho, 2001*). There have been two systems where such a condition-dependent fitness cost has been demonstrated: in the substrate-borne signaling of wolf spiders (*Hygrolycosa rubrofasciata*) and in the ornamented tail feathers of barn swallows (*Hirundo rustica*). In wolf spiders, individuals that are fed high-quantity diets maintain steady body mass and drum at greater rates compared to those fed a low-rationed diet (*Kotiaho, 2000*). Females prefer to mate with males that drum at higher rates (*Kotiaho et al., 1996*). However, drumming is energetically demanding and, sometimes, lethal (*Kotiaho, 2000*; *Mappes et al., 1996*). Males fed high-volume diets are better able to sustain and survive these costs compared to males on low-volume diets (*Kotiaho, 2000*).

In barn swallows, females prefer to mate with males that have long marginal tail feathers (*Møller, 1988*; *Møller, 1990*; *Møller, 1992*). However, long tail feathers hinder the aerodynamics of flight: Individuals with experimentally lengthened tail feathers catch smaller, lower-quality dipteran prey and are more likely to die. Meanwhile, those with experimentally shortened tail feathers catch larger, higher quality dipteran prey and were more likely to survive (*Moller and de Lope, 1994*). Individuals with naturally long tail feathers were best able to cope with experimental tail elongation, whereas those with naturally short tail feathers reaped the greatest survival boost from tail shortening, suggesting

that the cost of ornamented tail feathers disproportionately burdens males with naturally short tail feathers (*Moller and de Lope, 1994*).

It is not feasible to quantify such fitness costs in natural observations of snapping shrimp. They are prolific breeders, cryptic, and difficult to mark and recapture because they molt each month. The egg production tradeoffs are as close to a direct fitness cost as we could identify. Morphological tradeoffs, on the other hand, are more distant from fitness costs. However, it is a reasonable possibility that abdomen tradeoffs impact survival. For example, the primary mode of predator escape in many decapod crustaceans is the tailflip, during which individuals contract their abdomen to propel themselves backward (*Wiersma, 1947*). Tailflip velocity and acceleration in crayfish increase with abdomen length (*Hunyadi et al., 2020*). If the same holds in snapping shrimp, then the abdomen tradeoff that we found here could influence survival. However, future work is required to reach a definitive answer.

Some expenditures we documented did not differ with size; however, the overall fitness cost might still be size-dependent. For example, smaller females did not exhibit a weapon size tradeoff with the total number of eggs they produced. Even though the scaling slopes were invariant across the size range, small individuals might suffer a greater relative reduction in eggs and therefore a greater reduction in relative fitness. For example, reducing a 100-egg clutch by 10 would incur a 10% decrease, but reducing a 200-egg clutch by 10 would incur a 5%decrease. Compared to large and fecund individuals, then, smaller individuals might suffer a greater reduction in relative fitness than larger individuals despite a similar absolute tradeoff in egg production.

Critically, our findings rely entirely on observational data, and therefore, we cannot infer causation. In other taxa, it is possible to experimentally manipulate weapons to prevent them from developing. In beetles, for example, researchers can ablate imaginal disklike tissue in larvae before they form into horns (*Moczek and Nijhout, 2004*; *Simmons and Emlen, 2006*), and in coreid insects, researchers can induce permanent autotomy of hind-leg weapons (*Joseph et al., 2018*; *Miller et al., 2019*; *Somjee et al., 2018*). However, permanent manipulation of weaponry is not as straightforward in snapping shrimp as it is in terminally molting insects. Snapping shrimp molt every 16–20 days, and after autotomy, they regenerate their weapons over a series of molts. Specifically, snapping shrimp regenerate a new non-snapping claw at the site of autotomy and transform the contralateral claw into a snapping claw (*Cooney et al., 2017*; *Govind et al., 1986*; *Pereira et al., 2014*). It is not clear how weapon expenditures would change during regeneration; one might expect some expenditures, like maintenance, could decrease with a smaller, recovering claw, but other expenditures, like growth, could increase as the weapon grows and transforms (*Akhter et al., 2015*; *Bywater et al., 2014*; *Dinh, 2022*; *Pereira et al., 2014*; *Read and Govind, 1991*). Interestingly, in rare cases, wild-caught snapping shrimp can bear two snapping claws. This unusual arrangement can be permanently induced in the lab by removing the dactyl from a claw as it is transforming from a non-snapping claw to a snapping claw (*Read and Govind, 1997*). In future experiments, it would be interesting to test if this manipulation decreases investment into traits like abdomens and egg production.

Our methods also could not account for the genetic background of individuals, which could influence their weapon investment. Future experiments could use a siblings-based approach to control for these factors. Prior research has relied on breeding experiments, for example, to identify weapon-testes tradeoffs in horned scarab beetles (*O. taurus*) (*Moczek and Nijhout, 2004*) and to test the effects of diet on weapon development in the dung beetle *Onthophagus acuminatus* (*Emlen, 1997*). Indeed, it is possible to rear field-collected snapping shrimp eggs in captivity, though we have not been successful at inducing mating in the lab (*Harrison and Patek, 2023*). In the future, it would be interesting to use a siblings-based design with the experimental manipulations described above to test experimentally for resource allocation tradeoffs.

Despite the challenges of observational inference, the statistical approach of testing for residual-based tradeoffs pinpoints patterns that are consistent with resource allocation tradeoffs. The morphologies implicated in these trends are valuable candidates for future experiments as described above. Furthermore, the robust datasets that we are able to collect from field observations allowed us to test for size-dependence of these tradeoffs — a critical assumption of the handicap principle that has largely been omitted from such experiments. Future research could apply the statistical framework we lay out here to test for size dependence in tradeoffs in lab-based experiments as well.

Empirical evidence of fitness costs is elusive because fitness manifests from a mosaic of subtle expenditures. Some of these expenditures, like reproduction, are obviously correlated to fitness, while

others might have subtle yet meaningful effects. There is likely a smorgasbord of expenditures that we did not test for here, some of which are undetectable in purely observational work. For example, in other crustaceans, weapons hinder locomotion and reduce survival during predator escape (**Hunyadi et al., 2020**). These expenditures need to be identified through future experiments. Other expenditures might not be tractable through morphology, but through social interactions. In the paper wasp *Polistes dominulus*, for example, body size is correlated with pigment deposition in facial masks. Poor-condition wasps with facemasks manipulated to appear formidable experienced social costs via conspecific aggression (**Tibbetts and Dale, 2004**). The observational work we present here is a starting point to identify the fitness consequences of large weaponry. We encourage observations of behavior in naturalistic conditions and experiments that manipulate sexual traits to paint the entire mosaic of fitness-relevant expenditures of weaponry.

## Conclusion

The handicap principle suggests that individuals are plastic in their ability to signal at different levels, and they signal at the level that optimizes their cost-benefit difference (**Grafen, 1990a**; **Grafen, 1990b**; **Nur and Hasson, 1984**; **Zahavi, 1977**). This hypothesis requires costs or benefits that differ between individuals. However, the debate and acceptance of this principle have relied more on theory and less on empirical evidence (**Penn and Számadó, 2020**). We showed through field observations that size-dependent expenditures can ensure signal reliability through morphological and reproductive tradeoffs. Furthermore, we co-opted the same logic of differential costs and benefits to show that large weapons are particularly beneficial to males and particularly burdensome to females. These sex-specific implications of weaponry on reproduction could underlie sex and seasonal differences in costly trait expression.

# Materials and methods
## Animal collection

In total, we collected 677 *Alpheus heterochaelis* snapping shrimp from Beaufort, North Carolina, USA (NCDENR Scientific and Education permit # 707075 to Duke University Marine Laboratory). We measured each individual and tested for a tradeoff between the abdomen and snapping claw size (see *Morphological tradeoff* and *Seasonal trends* sections below). Subsets of these same *Alpheus heterochaelis* individuals were used in the remaining analyses: We used 76 individuals to test for kinematic tradeoffs (see *Kinematics* section), 37 egg-bearing females to test for reproductive tradeoffs (see *Reproductive tradeoffs* section), and 486 individuals to test for pairing benefits (see *Pairing* section). Finally, we captured 45 *Alpheus estuariensis* individuals from the same site and 53 *Alpheus angulosus* individuals from Beaufort, South Carolina, USA, and we tested whether morphological tradeoffs also arose in these species. No ethical permits were required.

We collected *A. heterochaelis* and *A. estuariensis* once per month during the spring tide from July to October 2020 and February to August 2021. We collected *A. angulosus* during one trip in March 2019. We found snapping shrimp in oyster reefs at low tide by flipping oyster clusters and excavating several centimeters of mud. We located individuals through turbid waters by scanning for antennae sweeping the water surface. We designated two shrimp as a male-female pair if they occupied the same tidepool underneath an oyster clump, and we acquired pairing data for 486 *Alpheus heterochaelis* individuals. We also noted whether individuals were caught during the breeding season. We considered breeding season as a binary variable. If any female was found holding eggs, then the collection date was considered the breeding season. The breeding season occurred between April and October, and no eggs were found during February and March collections. The months of breeding resemble those seen in *A. angulosus* populations in Charleston, South Carolina, USA (**Heuring and Hughes, 2019**). Temperatures in nearby waters were colder during the non-breeding season, fluctuating between 8 and 14°C, whereas breeding season temperatures fluctuated between 18–30°C (NOAA Station 8656483, Beaufort, Duke Marine Lab, North Carolina, USA).

For all three species, we measured each individual's carapace length, abdomen length, rostrum-to-telson length, and snapping claw length using digital calipers (resolution +/−0.02 mm, Husky Tools, Atlanta, Georgia, USA) (see *Supplementary file 1*, Figure 1, 2). We built log-log scaling relationships for snapping claws, and abdomen length as a function of rostrum-to-telson length, sex, and

their interaction. Abdomen length and carapace length both contribute to rostrum-to-telson length, but we used each of the three metrics in separate analyses because the existing literature indicates that each metric predicts different biologically relevant functions. For example, carapace length is the best-known predictor of resource-holding potential (*Dinh et al., 2020*; *Dinh and Patek, 2023*), abdomen length in other benthic decapods predicts predator escape velocity (*Hunyadi et al., 2020*), and rostrum-to-telson length predicts egg production (*Knowlton, 1980*).

## Statistical analysis

All statistical analyses were conducted using, R version 4.1.1, RStudio version 1.4.1717, and the tidyverse suite of R packages (*R Development Core Team, 2018*; *RStudioTeam, 2021*; *Wickham et al., 2019*).

## Morphological tradeoffs

For each species, we hypothesized that growing a larger snapping claw would coincide with reduced abdomen size. We tested this relationship by calculating the residuals from the log-log abdomen and snapping claw scaling relationships defined above, where positive residuals indicate a larger abdomen or snapping claw than predicted by the scaling relationship. To test for a morphological tradeoff, we built regressions using abdomen residuals as the response variable and snapping claw residuals, sex, and their interaction as the explanatory variables. We repeated this analysis for *A. heterochaelis*, *A. angulosus*, and *A. estuariensis*.

Then, we tested whether the slopes of the tradeoff depended on quality. Here and throughout the rest of the paper, we used carapace length as a measure of quality because it is the best-known proxy for resource-holding potential and a reliable predictor of dominance and subordinance in dyadic contests (*Dinh et al., 2020*). We hypothesized that the slope of the tradeoff would increase as carapace length decreased. To test this, we standardized carapace length so that the mean was zero and each increment of one represents an increase of one standard deviation. We built a regression with abdomen residual as the response variable and snapping claw residual, standardized carapace length, and their interaction as the explanatory variable. We performed this analysis only for *A. heterochaelis*, the species for which we had the greatest sample size and statistical power. We predicted a negative coefficient for the interaction, meaning that the tradeoff slope would approach zero as the carapace length increased.

## Kinematics

We reanalyzed data from *Dinh, 2022* to test if exaggerated weapons reduced weapon performance in *A. heterochaelis*. We recorded high-speed videos with synchronous pressure measurements from 10 snaps each in 76 individuals. We measured the average angular velocity, cavitation bubble duration, and peak-to-peak sound pressure level of each snap. Details about recording setup, equipment, and performance metrics are provided in *Dinh, 2022*. In brief, we calculated average angular velocity as the angle change between the dactyl and the propodus during closure divided by the duration of closure (*Kagaya and Patek, 2016*). Then, we calculated cavitation bubble duration as the duration between the initiation of cavitation to the onset of initial bubble implosion. Finally, we calculated the peak-to-peak sound pressure level coincident with the cavitation bubble collapse.

In previous research, we showed that average angular velocity decreased as claw mass increased, whereas cavitation bubble duration and sound pressure level increased as claw mass increased (*Dinh and Patek, 2023*). Here, we tested if these relationships also depended on weapon residuals. We built three linear models that used either $\log_{10}$(average angular velocity), $\log_{10}$(bubble duration), or sound pressure level (a logarithmic measure of pressure) as the response variable. In each model, we used $\log_{10}$(claw mass) and weapon residual as explanatory variables. We built separate models for males and females. For each performance metric, we hypothesized that performance would decrease with high-residual snapping claws, and we, therefore, predicted a negative coefficient for snapping claw residuals.

## Reproductive tradeoffs

We collected 37 ovigerous *A. heterochaelis* females. We removed each egg clutch and photographed them. We only included eggs in the early stage of development when the egg yolk was barely

consumed and oblong deformation by the embryo was minimal. We counted the total number of eggs in each egg clutch and measured the estimated average egg volume using the Fiji distribution of ImageJ (version 2.0.0) (*Schindelin et al., 2012*). For each egg clutch, we measured the egg volume for 20 randomly selected eggs as $V_{egg} = \frac{1}{6}\pi d_{min}^2 d_{max}$, where $V_{egg}$ represents egg volume, $d_{min}$ represents the minimum egg diameter, and $d_{max}$ represents the maximum egg diameter (*Kuris, 1990*). We then calculated the average egg volume as the mean volume of these 20 eggs. Finally, we calculated the total egg mass volume as the egg count multiplied by the average egg volume.

Egg count and egg mass volume increased as carapace length increased. Therefore, we regressed egg count and egg mass volume against carapace length and calculated egg count residuals and egg mass volume residuals from the scaling relationship. These residuals reflect investment into eggs, where more positive residuals indicate greater investment and more negative residuals indicate less investment. We did not use residual analysis for average egg volume because it did not scale with carapace length. To test for reproductive tradeoffs between eggs and weapons, we built three linear regressions that used either egg count residual, average egg volume, or egg mass volume residual as the response variable. All models included snapping claw residual as the sole explanatory variable. We predicted a negative relationship that reflected a reproductive tradeoff.

Then, to test if female snapping shrimp with smaller carapace lengths faced steeper tradeoffs, we added carapace length and its interaction with snapping claw residual to each of the models. If smaller individuals pay steeper expenditures, then the interaction should be positive: The negative relationship between egg properties and snapping claw residuals would taper to zero as carapace length increases.

## Pairing

We used *t*-tests to determine if paired individuals had greater weapon residuals than unpaired individuals. The response variable was weapon residual, and the explanatory variable was a binary variable of paired status, where one represents a paired individual and zero represents an unpaired individual. We performed separate tests for each sex.

Similarly, to test if greater snapping claw residuals increased the probability of pairing, we built a binomial generalized linear model with pairing status (1=paired, 0=unpaired) as the response variable. The explanatory variables were carapace length and snapping claw residual. We built models for each sex separately.

Then, we tested if individuals with greater weapon residuals paired with larger mates. We calculated the relative size of pair mates as $1 - \frac{rostrum-to-telson\ length_{focal}}{rostrum-to-telson\ length_{pairmate}}$ such that more positive values mean that pair mates are larger than focal individuals, and 0 means that individuals are equally sized. We used rostrum-to-telson length here because males and females form size-assortative pairs based on body length (*Mathews, 2002b*; *Nolan and Salmon, 1970*). We built a linear model with the relative size of pair mates as the response variable and the snapping claw residual of the focal individual as the explanatory variable. We repeated this analysis using either males or females as the focal individuals and the opposite sex as the pair mate. We predicted a positive relationship if individuals with greater weapon residuals attracted or maintained relatively larger pair mates.

## Seasonal trends

We tested if reproductive tradeoffs manifested in seasonal fluctuations in morphology between breeding and non-breeding seasons in *Alpheus heterochaelis*. We performed *t*-tests to compare (1) abdomen residuals and (2) snapping claw residuals using the breeding season as the explanatory variable (1=breeding season, 0=non-breeding season). The breeding season lasted from April to October when we found ovigerous female snapping shrimp. February and March collections were considered the nonbreeding season because we collected no ovigerous females. We performed separate t-tests for each sex in *Alpheus heterochaelis*. We predicted that snapping claw residuals would be elevated during the breeding season for males but not females, and that shift would coincide with a reduction in abdomen residuals. Then, to test if the scaling slope of the snapping claw changed between seasons, we built a linear model for each sex with $log_{10}$(snapping claw length) as the response variable and $log_{10}$(rostrum-to-telson length), breeding season, and their interaction as the predictor variables. A significant interaction term would indicate a seasonal allometric shift. If the interaction term was

nonsignificant, we removed it from the model to test if there was an overall shift in weapon investment without a change in slope across breeding and non-breeding seasons.

## Acknowledgements

Thanks to Ben Schelling and Jacob Harrison for assisting with field collection. Thanks also to the Duke University Marine Laboratory for providing facilities and administrative support. This research was funded by the Duke University Biology Department Grant-in-Aid and the NSF Graduate Research Fellowship to JPD and NSF IOS 2019323 to SNP.

## Additional information

### Funding

| Funder | Grant reference number | Author |
|---|---|---|
| National Science Foundation | IOS 2019323 | SN Patek |
| National Science Foundation | DGE 2139754 | Jason P Dinh |

The funders had no role in study design, data collection and interpretation, or the decision to submit the work for publication.

### Author contributions

Jason P Dinh, Conceptualization, Data curation, Software, Formal analysis, Funding acquisition, Investigation, Visualization, Methodology, Writing – original draft, Project administration, Writing – review and editing; SN Patek, Resources, Supervision, Funding acquisition, Methodology, Project administration, Writing – review and editing

### Author ORCIDs

Jason P Dinh ⓘ http://orcid.org/0000-0001-6471-9047
SN Patek ⓘ http://orcid.org/0000-0001-9738-882X

### Decision letter and Author response

Decision letter https://doi.org/10.7554/eLife.84589.sa1
Author response https://doi.org/10.7554/eLife.84589.sa2

## Additional files

### Supplementary files
• MDAR checklist

• Supplementary file 1. Abdomen length and snapping claw length scaled positively with rostrum-to-telson length.

### Data availability

Data, metadata, and code are available on Dryad: https://doi.org/10.5061/dryad.qz612jmkf.

The following dataset was generated:

| Author(s) | Year | Dataset title | Dataset URL | Database and Identifier |
|---|---|---|---|---|
| Dinh JP, Patek SN | 2022 | Tradeoffs and benefits explain scaling, sex differences, and seasonal oscillations in the remarkable weapons of snapping shrimp (Alpheus spp.) | https://dx.doi.org/10.5061/dryad.qz612jmkf | Dryad Digital Repository, 10.5061/dryad.qz612jmkf |

The following previously published dataset was used:

| Author(s) | Year | Dataset title | Dataset URL | Database and Identifier |
|---|---|---|---|---|
| Dinh JP, Patek S | 2022 | Weapon performance and contest assessment strategies of the cavitating snaps in snapping shrimp | https://dx.doi.org/10.5061/dryad.qz612jmjx | Dryad Digital Repository, 10.5061/dryad.qz612jmjx |

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
