## [Editor Report]

This study on snapping shrimp morphological weaponry presents important findings on trade-offs in investment in costly weaponry traits as related to body size and reproduction. Convincing evidence is based on the collection of an exceptional number of fields samples, the inclusion of three shrimp species, and the measurement of numerous morphological and behavioral traits. The evidence shows that there are size-dependent trade-offs. Further, males and females differ in weapon investment, as weapons are especially beneficial to males but especially expensive for females. The findings will be of broad interest to evolutionary biologists and researchers working in the field of animal behavior.

---

## [Decision Letter]

**Decision letter after peer review:**

Thank you for submitting your article "Tradeoffs and Benefits Explain Scaling, Sex Differences, and Seasonal Oscillations in the Remarkable Weapons of Snapping Shrimp (Alpheus spp.)" for consideration by *eLife*. Your article has been reviewed by 2 peer reviewers, including Lauren A O'Connell as the Reviewing Editor and Reviewer #1, and the evaluation has been overseen by George Perry as the Senior Editor. The following individual involved in review of your submission has agreed to reveal their identity: Christine Whitney Miller (Reviewer #2).

Essential revisions:

1) Please consider a rewrite to tighten up the manuscript to make the paper clearer to the non specialist reader. The writing is meandering at times and the discussion could be restructured to link these findings to the broader literature.

2) In the discussion, please include more text about concrete costs and benefits as well as the pros and cons of the approach.

*Reviewer #1 (Recommendations for the authors):*

I commend the authors for all the natural observations done in these three shrimp species – this was a ton of work that allows for a robust testing of some evolutionary theory.

The introduction is a bit meandering. This is more a matter of style, but I would find it easier to read if the testing and predictions paragraphs in the intro were moved into the results to better set up the narrative and allow one to connect each result to a main idea. I get that this framework makes sense when a single idea or hypothesis is being tested, but there are many and by the time I got to the results, I had forgotten the main idea being tested. Sprinkling them throughout the results would prevent this from happening.

The discussion writing needs to be cleaned up a bit and is heavily focused on repeating the results and talking about shrimp. Starting in paragraph two of the discussion, there are 2-3 sentence mini-paragraphs that repeat the results. I don't think this are necessary. I would have liked to have seen interpretations in the light of other literature sooner rather than a repetition of what I had just read in the results. Consider filling these paragraphs out with comparisons to other studies or removing them/ incorporating them into the writing better. The first really great paragraph in the discussion that does this is on line 462 – 7 paragraphs in. There is very limited writing comparing these results to other taxa – doing this would improve the reach of your findings beyond something cool in snapping shrimp to a general principle of trait evolution that you suggest.

Species differences – The allometric slop of the snapping claw scaling differed between sexes in two of three species – why is that?

*Reviewer #2 (Recommendations for the authors):*

The current version of the manuscript would benefit from a discussion of the pros and cons of their approach of using residuals versus other approaches to measure resource allocation trade-offs. This version also needs more information on the life-history of these shrimp, including lifespan and information on molting. Further discussion is also needed on the incorporation of carapace length and rostrum-to-telson length and their relation to each other. Together, this information will help the general reader better interpret the results.

More details and other notes:

1. Title: Consider removing "benefits" from the title because it just makes the title more complex without providing much information.

2. Contractions are used throughout, which is not formal scientific writing (e.g., Lines 78 & 460).

3. Line 16. To be a little more exacting, these species show patterns consistent with resource allocation trade-offs rather than just "exhibit resource allocation trade-offs". See below -- more discussion on the inference provided by different methods to measure trade-offs will also be helpful.

4. Line 27. "Especially burdensome to females", yet the authors have not tested for benefits to females of having weapons, which might be profound.

5. Lines 78-79. Other than the use of the contraction, this is an excellent way to set up the patterns – this manuscript has great writing throughout.

6. The paper in its current form is excellent, however there are some missing bits that are essential for the general reader to be able to interpret the quality of the paper and make sense of the patterns:

a. I find it essential that the authors address the pros and cons of their approach to measuring resource allocation trade-offs, and their approach is just one of many ways of looking at whether or not such trade-offs exist and the extent of the trade-offs. This is the #1 omission of the manuscript in its current form. What can the use of residuals say, what are the possible problems with this approach, and what are its strengths? Are results from this approach generally consistent with other approaches used? This approach does not account for genetics, nor is this an experimental method. Make sure to include citations throughout this discussion.

b. What is the lifespan of these invertebrates? The authors sampled two species monthly over a several month period in 2020 and 2021. How does this sampling overlap with their lifespan and molting cycles? When do they molt? How often do they molt? And, the authors mentioned that they change in their body dimensions during the reproductive season. Indeed, what is possible when they do molt – how much can they change? Most people reading this study will not have familiarity with marine invertebrates and especially these species.

c. Line 152. Both carapace length and rostrum-to-telson length are measured. The authors state that carapace length is a measure of quality. Though not explicitly stated here, it seems that rostrum-to-telson length is the measure they use of body size. This should be explicitly stated and why this metric is chosen. It would also be helpful to describe why carapace length should be the measure of quality, while the other is the size metric. How do they scale with each other?

7. Lines 86-88. This is a conservative and reasonable approach, and yet "costs" is used commonly throughout the manuscript (see Discussion section). It would be valuable to come back to this statement in the discussion and make sure that the terms used throughout are consistent.

8. Lines 427-430. How precise are the measurements? Could the difference be relatively small and hard to detect with your current methods?

9. Lines 477-487 More discussion of concrete costs and benefits would be helpful. In particular, there is mention in the introduction that females use their weapons for resource defense, but it is not discussed again.

---

## [Author Response]

Essential revisions:1) Please consider a rewrite to tighten up the manuscript to make the paper clearer to the non specialist reader. The writing is meandering at times and the discussion could be restructured to link these findings to the broader literature.

We’ve rewritten the manuscript as suggested in the comments below. See specific comments below for our responses.

2) In the discussion, please include more text about concrete costs and benefits as well as the pros and cons of the approach.

We’ve included an extensive discussion about the concrete costs and benefits as well as the pros and cons of our observational approach. See specific comments below for our responses.

Reviewer #1 (Recommendations for the authors):I commend the authors for all the natural observations done in these three shrimp species – this was a ton of work that allows for a robust testing of some evolutionary theory.The introduction is a bit meandering. This is more a matter of style, but I would find it easier to read if the testing and predictions paragraphs in the intro were moved into the results to better set up the narrative and allow one to connect each result to a main idea. I get that this framework makes sense when a single idea or hypothesis is being tested, but there are many and by the time I got to the results, I had forgotten the main idea being tested. Sprinkling them throughout the results would prevent this from happening.

We have now moved these paragraphs to the results, as suggested.

The discussion writing needs to be cleaned up a bit and is heavily focused on repeating the results and talking about shrimp. Starting in paragraph two of the discussion, there are 2-3 sentence mini-paragraphs that repeat the results. I don't think this are necessary. I would have liked to have seen interpretations in the light of other literature sooner rather than a repetition of what I had just read in the results. Consider filling these paragraphs out with comparisons to other studies or removing them/ incorporating them into the writing better. The first really great paragraph in the discussion that does this is on line 462 – 7 paragraphs in. There is very limited writing comparing these results to other taxa – doing this would improve the reach of your findings beyond something cool in snapping shrimp to a general principle of trait evolution that you suggest.

We have now filled in these paragraphs with relevant findings from the existing literature and other taxa, as suggested.

Species differences – The allometric slop of the snapping claw scaling differed between sexes in two of three species – why is that?

We are not entirely sure. It’s possible that this was just a sample size issue. The species where we did not detect a statistically significant difference in allometric slope also had the smallest sample size. We added this caveat in lines 125-126.

Reviewer #2 (Recommendations for the authors):The current version of the manuscript would benefit from a discussion of the pros and cons of their approach of using residuals versus other approaches to measure resource allocation trade-offs.

We have now included an extensive discussion of the pros, cons, and future directions of our method in lines 637-676.

This version also needs more information on the life-history of these shrimp, including lifespan and information on molting.

We have now included information on life history and molting in lines 62-74 and lines 642-655.

Further discussion is also needed on the incorporation of carapace length and rostrum-to-telson length and their relation to each other. Together, this information will help the general reader better interpret the results.

The latest manuscript includes this discussion in lines 743-748.

More details and other notes:1. Title: Consider removing "benefits" from the title because it just makes the title more complex without providing much information.

Done.

2. Contractions are used throughout, which is not formal scientific writing (e.g., Lines 78 & 460).

Done.

3. Line 16. To be a little more exacting, these species show patterns consistent with resource allocation trade-offs rather than just "exhibit resource allocation trade-offs". See below -- more discussion on the inference provided by different methods to measure trade-offs will also be helpful.

Done.

4. Line 27. "Especially burdensome to females", yet the authors have not tested for benefits to females of having weapons, which might be profound.

Females certainly benefit from bearing large weapons. Large weapons have greater offensive capacity and provide competitive advantage in intrasexual combat for both sexes, for example (Dinh et al., 2020; Dinh & Patek, 2023).

The key sex difference we identify in this paper is in pairing benefits. We tested for this in both males and females, but only males exhibited this benefit. That’s why we conclude that weapons are especially beneficial to males—not exclusively beneficial to males.

5. Lines 78-79. Other than the use of the contraction, this is an excellent way to set up the patterns – this manuscript has great writing throughout.

Thank you!

6. The paper in its current form is excellent, however there are some missing bits that are essential for the general reader to be able to interpret the quality of the paper and make sense of the patterns:a. I find it essential that the authors address the pros and cons of their approach to measuring resource allocation trade-offs, and their approach is just one of many ways of looking at whether or not such trade-offs exist and the extent of the trade-offs. This is the #1 omission of the manuscript in its current form. What can the use of residuals say, what are the possible problems with this approach, and what are its strengths? Are results from this approach generally consistent with other approaches used? This approach does not account for genetics, nor is this an experimental method. Make sure to include citations throughout this discussion.

We’ve now included a discussion about the pros and cons of the observational residual approach. We’ve also discussed how the approach can be integrated with alternative experimental approaches in the future (lines 637-676).

b. What is the lifespan of these invertebrates? The authors sampled two species monthly over a several month period in 2020 and 2021. How does this sampling overlap with their lifespan and molting cycles? When do they molt? How often do they molt? And, the authors mentioned that they change in their body dimensions during the reproductive season. Indeed, what is possible when they do molt – how much can they change? Most people reading this study will not have familiarity with marine invertebrates and especially these species.

They molt year round every 16 to 20 days, on average. Indeed, they can grow and alter their proportions — and even regenerate and transform claws — between molts.

We’ve now included these natural history details in lines 62-74 and lines 642-655.

c. Line 152. Both carapace length and rostrum-to-telson length are measured. The authors state that carapace length is a measure of quality. Though not explicitly stated here, it seems that rostrum-to-telson length is the measure they use of body size. This should be explicitly stated and why this metric is chosen. It would also be helpful to describe why carapace length should be the measure of quality, while the other is the size metric. How do they scale with each other?

We’ve now included a discussion of rostrum-to-telson length, carapace length, and abdomen length in lines 743-748. They are all correlated, but previous research has implicated each one in different important biological processes that we are interested in.

7. Lines 86-88. This is a conservative and reasonable approach, and yet "costs" is used commonly throughout the manuscript (see Discussion section). It would be valuable to come back to this statement in the discussion and make sure that the terms used throughout are consistent.

We use “costs” when speaking more broadly about the theory and the broader literature, but we used “expenditure” or “tradeoff” when talking specifically about our results. We double-checked our terminology throughout and made all necessary corrections.

8. Lines 427-430. How precise are the measurements? Could the difference be relatively small and hard to detect with your current methods?

Our measurements were extremely precise. We describe our methods and equipment in Dinh & Patek (2023), but in short, we used a high-speed camera that recorded at 100,000 frames per second and a calibrated hydrophone that recorded 1,000,000 samples per second. That provided 0.01 millisecond resolution in kinematic measurements and microsecond resolution in pressure measurements.

9. Lines 477-487 More discussion of concrete costs and benefits would be helpful. In particular, there is mention in the introduction that females use their weapons for resource defense, but it is not discussed again.

We have now included a paragraph of concrete costs and benefits in lines 573-591.